# Asymmetric α-benzylation of cyclic ketones enabled by concurrent chemical aldol condensation and biocatalytic reduction

Yunting Liu [1], Teng Ma [1], Zhongxu Guo[1], Liya Zhou[1], Guanhua Liu[1], Ying He[1], Li Ma[1], Jing Gao[1], Jing Bai[2], Frank Hollmann [3]✉ & Yanjun Jiang [1]✉

Chemoenzymatic cascade catalysis has emerged as a revolutionary tool for streamlining traditional retrosynthetic disconnections, creating new possibilities for the asymmetric synthesis of valuable chiral compounds. Here we construct a one-pot concurrent chemoenzymatic cascade by integrating organobismuth-catalyzed aldol condensation with ene-reductase (ER)-catalyzed enantioselective reduction, enabling the formal asymmetric α-benzylation of cyclic ketones. To achieve this, we develop a pair of enantio-complementary ERs capable of reducing α-arylidene cyclic ketones, lactams, and lactones. Our engineered mutants exhibit significantly higher activity, up to 37-fold, and broader substrate specificity compared to the parent enzyme. The key to success is due to the well-tuned hydride attack distance/angle and, more importantly, to the synergistic proton-delivery triade of Tyr28-Tyr69-Tyr169. Molecular docking and density functional theory (DFT) studies provide important insights into the bioreduction mechanisms. Furthermore, we demonstrate the synthetic utility of the best mutants in the asymmetric synthesis of several key chiral synthons.

The benzyl motif is a frequent pharmacophore, particularly in many anti-cancer compounds[1]. Chiral α-benzyl cyclic carbonyl compounds, such as ketones, lactams, and lactones, are valuable building blocks for the synthesis of pharmaceuticals, biologically active molecules, and natural products (Fig. 1a)[2–5]. Given the importance of this motif, various enantioselective transformations have been developed to access these chiral compounds, including Ir-catalyzed asymmetric hydrogenation[6–10], Rh-catalyzed allylic alcohol isomerization[11], Pd-catalyzed decarboxylative protonation[12,13], and organocatalyzed enol derivative protonation[14]. While these approaches are effective, they still often include harsh reaction conditions, the use of stoichiometric reagents and volatile organic solvents. Also, racemization of the products of interest represents an issue frequently observed. In recent years, biocatalysis has emerged as an environmentally friendly alternative for efficient, selective, and sustainable synthesis[15–18].

Unfortunately, no biocatalytic transformations have been reported to access the above-mentioned molecules.

Chemoenzymatic cascades, which combine the reaction diversity of chemical catalysts with the unparalleled selectivity of enzymes, have created numerous possibilities and expanded the repertoire of these two catalytic disciplines[19–23]. From an application point of view, one-pot concurrent chemoenzymatic cascades are particularly attractive as they allow the direct conversion of simple substrates to desired chiral products through multi-step reactions without intermediate operations. A common design approach is to couple chemocatalytic C-C bond formation with enzymatic stereocenter generation at carbon atoms distal to one another. The archetypal example for this approach, a chemoenzymatic cascade for enantioselective synthesis of biaryl alcohols was developed by Gröger and coworkers (Fig. 1b)[24]. Although uncommon, enzymatic generation of a stereocenter at one of the

[1]School of Chemical Engineering and Technology, Hebei University of Technology, Tianjin 300130, China. [2]College of Food Science and Biology, Hebei University of Science & Technology, Shijiazhuang 050018, China. [3]Department of Biotechnology, Delft University of Technology, 2629 HZ Delft, The Netherlands. ✉e-mail: f.hollmann@tudelft.nl; yanjunjiang@hebut.edu.cn

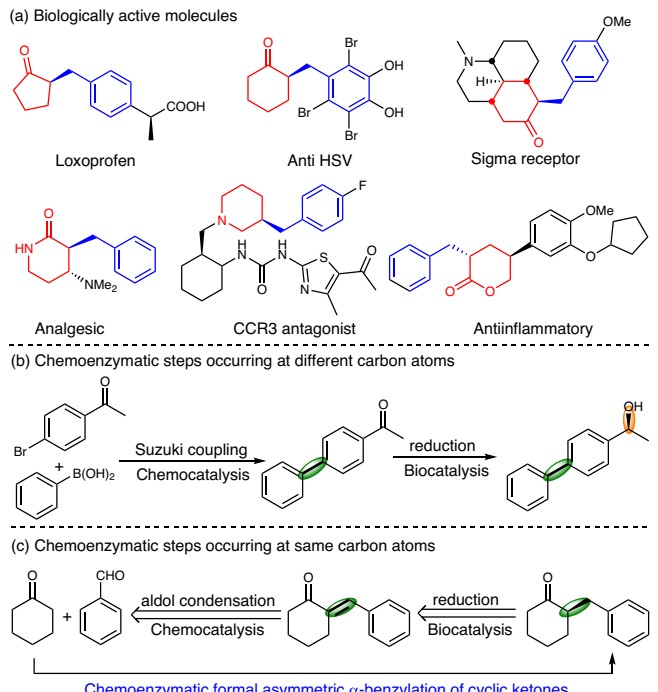

**Fig. 1 | Design of a chemoenzymatic cascade for the synthesis of chiral α-benzyl cyclic ketones. a** Selected examples of biologically active molecules bearing α-benzyl (blue) and cyclic carbonyl (red) groups. **b** Chemoenzymatic cascade with bond formation (green-shaded bond) and stereocenter generation (orange-shaded bond) occurring at different carbon atoms. **c** Chemoenzymatic cascade with bond formation and stereocenter generation occurring at same carbon atoms for formal asymmetric α-benzylation of cyclic ketones.

chemically reacting carbons offers a new strategy for synthetic pathway design[25–28]. For example, Zhao et al. developed a chemoenzymatic cascade for formal asymmetric C-C bond formation by combining metal-catalyzed construction of C=C bond and enzymatic reduction of the newly formed C=C bond[28]. The group around Gröger combined an organocatalytic aldol condensation reaction with an enzymatic (yet non-stereoselective) C=C bond reduction[29]. The incompatibility of the reagents and reaction conditions, however, forced the authors to perform the reaction in individual steps including intermediate product isolation. After performing a retrosynthetic analysis[30], we envisioned chiral α-benzyl cyclic carbonyl compounds originating from the enzymatic reduction of aryl-substituted exocyclic C=C double bonds, with the exocyclic substrates stemming from the aldol condensation of simple cyclic carbonyls and arylaldehydes (Fig. 1c). Hence, we envisioned a formal asymmetric α-benzylation of cyclic carbonyls.

Yin et al. established an organobismuth complex/aliphatic amine system for the synthesis of exocyclic α,β-unsaturated compounds via aldol condensation under mild reaction conditions[31], providing a potential chemocatalysis module for the proposed chemoenzymatic cascade. However, for the envisioned enzymatic module, *i.e.*, the bioreduction of aryl-substituted exocyclic C=C double bonds, no precedents are known yet. We, therefore, envisioned ene-reductases (ERs), catalyzing the stereoselective reduction of conjugated C=C-double bonds[32–38], to fill this gap. The substrate scope of ERs is generally limited to acyclic and endocyclic compounds, with exocyclic substrates remaining highly challenging[39–41]. Particularly, to the best of our knowledge, there is currently no enzymatic reduction of α-arylidene cyclic carbonyl compounds known. In addition, enantio-complementary ER pairs are rare because almost all wild-type (wt)-ERs displayed the same natural stereospecificity, making the synthesis of two mirror-image molecules very difficult. Directed evolution approaches have been applied to discover new ER members with

improved performances related to reactivity, substrate specificity and stereospecificity[42–46], providing possibilities of identifying productive ERs for the reduction of exocyclic C=C double bonds.

In this context, we herein report the engineering of an ER (YqjM from *Bacillus subtilis*) through a mechanism-guided directed evolution approach, discovering a pair of enantiocomplementary YqjM mutants for the reduction of exocyclic α,β-unsaturated carbonyl compounds including ketones, lactams, and lactones. A one-pot concurrent chemoenzymatic cascade for formal asymmetric α-benzylation of cyclic ketones was developed by combining organobismuth-catalyzed aldol condensation and YqjM-catalyzed reduction, producing enantiopure α-benzyl cyclic ketones in high yields (up to 98%) and excellent enantioselectivities (up to 99% ee). The applicability of the YqjM mutants was further confirmed in the synthesis of key intermediates of pharmaceutically relevant molecules.

## Results and discussion

### Molecular docking analysis of substrate binding poses/energies

To identify new ERs with specificity toward the challenging exocyclic substrates, the structurally well-characterized YqjM[47] from *Bacillus subtilis* was selected for activity investigation and subsequent protein design. Unsurprisingly, wt-YqjM showed extremely low specific activity (13.2 U/g_{protein}) toward (*E*)−2-benzylidenecyclopentan-1-one (**1a**), an exocyclic substrate, with only 3.5% conversion, despite the high stereoselectivity (96% ee) of the product (*S*)-**1ap**. To analyze the interactions between YqjM and **1a**, we performed in silico molecular docking. The productive binding poses should meet the three criteria: (1) formation of hydrogen bonds between His 164/167 with the substrate carbonyl oxygen to activate the C=C double bond, (2) the distance between the flavin mononucleotide (FMN) N5 atom and the alkene Cβ should be shorter than 4.1 Å, and (3) the hydride attack angle (N10...N5...Cβ) ranges between 80° and 120°[48]. The docking mode of YqjM with **1a** showed a poor binding pose with hydride attack distance and angle of 4.3 Å and 68°, respectively (Supplementary Fig. 1), which may account for its low activity.

We speculated that the Ile69, situated in a loop region very close to the active center, may play an important role in tuning the substrate binding poses. To validate this, the binding energy between YqjM mutants with saturation mutagenesis at residue 69 and **1a** was calculated by molecular docking approach. The results evidenced that the properties of amino acid at position 69 significantly affected the substrate affinity (Supplementary Table 3). For example, the larger I69W showed a higher binding energy (−0.48 kJ/mol) compared to that of the wt parent (−1.13 kJ/mol), while the smaller I69V, I69A and I69G mutants resulted in significantly reduced binding energy. The latter can be attributed to the lower steric demand of the smaller amino acids (Supplementary Fig. 2), facilitating the entry of substrates in a more flexible binding mode. Unexpectedly, the sulfur-containing cysteine and methionine also increased the binding energy. To experimentally validate the simulation results, a saturation mutagenesis library at position 69 was constructed (Table 1 and Supplementary Table 4). As anticipated, the smaller I69V, I69A, and I69G mutants displayed enhanced activity, while the larger I69W and the sulfur-containing mutants (I69C and I69M) exhibited lower activity. Surprisingly, mutation I69Y, albeit not small, presented the lowest binding energy of −2.58 kJ/mol and the highest activity of 125.6 U/g_{protein}. The reasons for this unexpected observation were further explored.

### Dual role of Tyr69 in the catalytic mechanism

Molecular docking revealed that I69Y created a narrow and shallow cavity positioning the substrate phenyl group almost perpendicular to the FMN plane (Fig. 2a, b) via an edge-to-face π-π interaction between the two aromatic systems from **1a** and Tyr69. Such interactions are well-documented in the literature[49,50]. This brings the substrate C=C double bond closer to the FMN N5, shortening the hydride attack

distance to 3.4 Å and improving the hydride attack angle to 82° (Supplementary Fig. 3). Additionally, I69Y and I69F exhibited large differences in activity (125.6 vs 35.6 U/g$_{protein}$), despite their structural similarity (Supplementary Fig. 4), confirming the important role of the Tyr69 hydroxyl group for the protonation of the intermediate enolate after hydride transfer. Kohli and Massey as well as Pietruszka and coworkers previously emphasized the importance of the catalytic Tyr as proton relay[45,51]. Therefore, the mutants Y169F and I69Y/Y169F were created. While Y169F exhibited almost no catalytic activity, I69Y/Y169F still displayed moderate activity (53.5 U/g$_{protein}$). Thus, the newly introduced Tyr69 not only accommodated the substrate binding in a favorable mode, but also participated in proton delivery during catalysis.

## DFT calculations suggest different mechanisms for protonation of the intermediate enolate by Tyr169 and Tyr69

Tyrosine can protonate the substrate directly (path 1) or via a bridging water molecule (path 2)[52]. To gain a deeper insight into the protonation mechanism by Tyr169 and Tyr69, DFT calculations were performed using wt-YqjM and the mutant I69Y/Y169F (each containing only one

### Table 1 | Specific activity of YqjM mutants toward 1a

| Mutants | Mutations | Specific activity (U/g$_{protein}$)[a] | Conversion (%)[b] |
|---|---|---|---|
| M$_0$ | wt | 13.2 ± 0.5 | 3.5 |
| M1$_1$ | I69A | 45.2 ± 1.2 | 36.6 |
| M1$_2$ | I69G | 58.5 ± 1.4 | 42.5 |
| M1$_3$ | I69V | 33.4 ± 0.5 | 26.9 |
| M1$_4$ | I69Y | 125.6 ± 3.5 | 85 |
| M1$_5$ | I69F | 35.6 ± 1.3 | 38 |
| M1$_6$ | Y169F | ND[c] | ND[c] |
| M2$_1$ | I69Y/C26A | 353.2 ± 9.3 | 91 |
| M2$_2$ | I69Y/C26G | 486.5 ± 13.2 | 96 |
| M2$_3$ | I69Y/Y169F | 53.5 ± 1.5 | 43.5 |
| M3$_1$ | I69Y/C26G/Y169F | 121.9 ± 2.3 | 79 |
| M3$_2$ | I69Y/C26G/Y28F | 195.9 ± 5.5 | 86 |

[a] Specific activity was calculated using the conversion at 1 min. U refers to the activity unit expressed as micromoles of substrate converted per minute. Reaction conditions: **1a** (1.0 mM), NADPH (2.0 mM), and YqjM mutants (10 µM) in PBS (1.0 mL, 100 mM, pH 7.5), reacting at 30 °C for 2 min. Data are obtained from three independent experiments and expressed as the mean ± SEM.
[b] The reduction of **1a** (10 mM) using YqjM mutants (0.1 mM) and NADPH (20 mM) in PBS (5.0 mL, 100 mM, pH 7.5) at 30 °C for 24 h was used as the template reaction for evaluating the conversions, which were determined by chiral GC. Mean values from triplicates are presented. Source data are provided as a Source data file.
[c] ND means not detected.

of the above proton transfer sites) as model enzymes. The stationary point energies were obtained according to a previously reported method[53]. The DFT results strongly supported path 1 for wt-YqjM to transfer a proton directly (Supplementary Table 5) due to the proximity of Tyr 169 and the enolate (<4 Å) not leaving sufficient space to accommodate a bridging water molecule. In contrast in I69Y/Y169F, path 2 is suggested to be energetically most favorable (Supplementary Table 6), which might be due to that the distance between the Tyr169-OH and the enolate Cα is too far for efficient proton transfer. The two reaction schemes with the lowest overall energy barriers were shown in Supplementary Figs. 5, 6.

## Evolution of a synergistic proton-delivery triade (Tyr28-Tyr69-Tyr169)

Inspired by the mutation I69Y, we hypothesized that other tyrosine residues surrounding the substrate may further improve the catalytic activity via enhancing the proton delivery. For example, tyrosine at position 28 was also found in the vicinity of the substrate. However, considering the negligible activity of Y169F, it can be concluded that the Tyr28 in wt-YqjM has no proton delivery ability. A detailed inspection of the substrate pocket suggested the neighboring Cys26 blocking direct access of **1a** to Tyr28 (Supplementary Fig. 1). The long distance between the Cys-SH and the alkene Cα renders an involvement of Cys26 as proton source in the catalytic mechanism unlikely. Thus, we hypothesized that replacing Cys26 with the smaller Gly may help in positioning **1a** more favorably towards Tyr28 for proton transfer. To confirm this, we introduced the second mutation, C26G, and the activity of the mutant I69Y/C26G reached 486.5 U/g$_{protein}$, which was 4-fold higher than that of the mutant I69Y. Docking models revealed that these changes reduced the hydride attack distance to 2.5 Å and increased the hydride attack angle to 91° (Fig. 2c and Supplementary Fig. 7), further enhancing the hydride transfer step. Notably, when Tyr was mutated to Phe, the obtained mutant I69Y/C26G/Y28F has drastically-lowered activity (195.9 U/g$_{protein}$) compared to the mutant I69Y/C26G, despite the similar binding poses between these two YqjM mutants. These results also provided solid evidence for the proton delivery ability of Tyr28. Overall, a synergistic proton-delivery triade (the activated Tyr28, the newly introduced Tyr69, and the original Tyr169) was stepwise engineered in the best mutant I69Y/C26G.

## Kinetic analysis of YqjM mutants

Kinetic parameters of several representative mutants for reduction of **1a** and (E)−2-benzylidenecyclohexan-1-one (**2a**) were investigated (Table 2 and Supplementary Figs. 9–22). Compared to wt-YqjM, both I69G and I69Y exhibited increased $k_{cat}/K_M$ values, but with different incentives. Mutation I69G plays a more important role in affecting $K_M$,

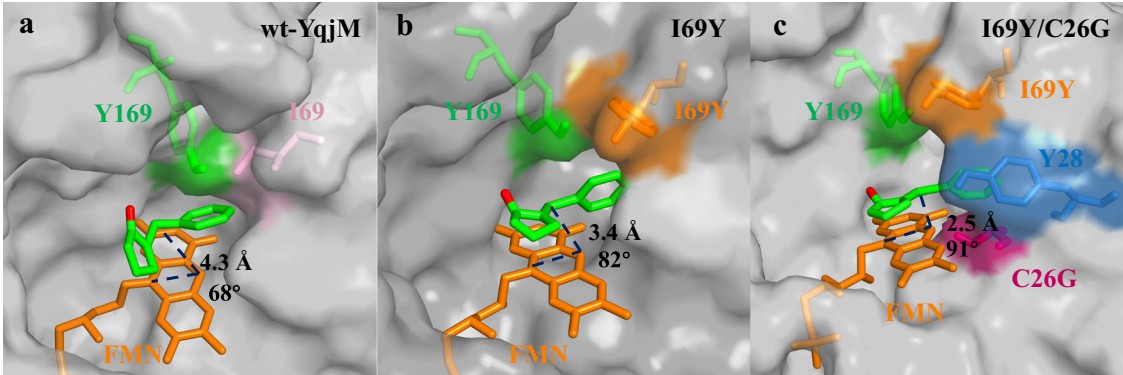

**Fig. 2 | Molecular docking results.** Models of the active pocket of the wt-YqjM (**a**), YqjM (I69Y) (**b**), and YqjM (I69Y/C26G) (**c**) containing substrate **1a**. The three residues, original Tyr169 (green), newly introduced Tyr69 (orange), and activated Tyr28 (blue), constitute a synergistic proton-delivery triade.

**Table 2 | Kinetic parameters of the YqjM mutants toward the substrates 1a and 2a.[a]**

| Mutants | Substrates | $K_M$ (mM) | $k_{cat}$ (s$^{-1}$) | $k_{cat}/K_M$ (s$^{-1}$ mM$^{-1}$) | Fold |
|---|---|---|---|---|---|
| wt | **1a** | 8.35 ± 0.2 | 0.23 ± 0.005 | 0.03 | 1 |
| I69G | | 3.56 ± 0.08 | 0.39 ± 0.012 | 0.11 | 4 |
| I69Y | | 4.25 ± 0.09 | 0.65 ± 0.015 | 0.15 | 5 |
| I69Y/C26A | | 2.65 ± 0.02 | 0.89 ± 0.023 | 0.34 | 11 |
| I69Y/C26G | | 1.21 ± 0.01 | 1.59 ± 0.03 | 1.31 | 44 |
| I69Y/C26G/Y169F | | 1.69 ± 0.01 | 0.56 ± 0.009 | 0.33 | 11 |
| I69Y/C26G/Y28F | | 2.21 ± 0.02 | 0.49 ± 0.012 | 0.22 | 7 |
| wt | **2a** | 7.98 ± 0.18 | 0.19 ± 0.009 | 0.02 | 1 |
| I69G | | 3.48 ± 0.11 | 0.43 ± 0.01 | 0.12 | 6 |
| I69Y | | 4.56 ± 0.12 | 0.87 ± 0.012 | 0.19 | 10 |
| I69Y/C26A | | 2.98 ± 0.1 | 1.12 ± 0.02 | 0.38 | 19 |
| I69Y/C26G | | 1.1 ± 0.03 | 1.86 ± 0.05 | 1.69 | 85 |
| I69Y/C26G/Y169F | | 1.75 ± 0.05 | 0.43 ± 0.01 | 0.25 | 13 |
| I69Y/C26G/Y28F | | 2.18 ± 0.04 | 0.53 ± 0.013 | 0.24 | 12 |

[a] Kinetic parameters were determined in PBS (100 mM, pH 7.5) containing 1.0 mM NADPH, 0-8 mM substrate, and 10 μM YqjM mutants, at 30 °C for 10 min. Experimental details see the Methods section. Source data are provided as a Source data file.

while I69Y has a more pronounced effect on $k_{cat}$. These results indicated that the I69G more favors the substrate affinity enhancement, while the I69Y favors the reaction rate improvement, which were attributed to the enlarged cavity of I69G and the additional proton-delivery site of I69Y, respectively. Combining the two factors, I69Y/C26G led to significant improvement in both substrate affinity and reaction rate, achieving 44- and 85-fold increases in $k_{cat}/K_M$ toward **1a** and **2a**, respectively. For I69Y/C26G/Y169F and I69Y/C26G/Y28F, the $k_{cat}$ decreased sharply due to the elimination of a proton-delivery site, while the $K_M$ increased only slightly in line with the minor change in the volume of the active pocket.

**Reshaping of the substrate pocket for enantioselectivity switch**
Having identified a highly active YqjM variant (I69Y/C26G) for the enantioselective reduction of exocyclic α,β-unsaturated ketones, we turned our attention to alter the enantioselectivity. Reanalyzing the binding model of **1a** in the YqjM active pocket (Supplementary Fig. 1), we found that the open cavity surrounded by the Cys26, Tyr28 and Ile69 amino acid residues favors accommodating the phenyl group of **1a**, thus generating (S)-stereoselective product. We hypothesized that limiting the space available in this cavity may force **1a** to bind in the opposite orientation and thereby invert the stereoselectivity of the reduction reaction. Hence, we targeted the sites Cys26, Tyr28, and Ile69. In contrast to our initial assumption, mutations with larger amino acids at positions 28 and 69 did not generally lead to the desired result. To our delight, however, YqjM-C26F displayed good activity (158.6 U/g) with high (R)-enantioselectivity (95% ee). Molecular docking results revealed that this mutation led to a productive binding pose with hydride attack distance and angle of 3.6 Å and 82°, respectively (Supplementary Fig. 8). Interestingly, the C26W mutation with larger Trp failed to increase the enantioselectivity (89% ee R) and caused a slight decrease in enzyme activity.

**Establishment of the enzymatic reduction conditions**
With the (S)-stereoselective YqjM (I69Y/C26G) and (R)-stereoselective YqjM (C26F) at hand, the enzymatic properties were firstly explored. As shown in Supplementary Figs. 23–30, these two mutants exhibited the best activity at pH 8.0 and 35 °C, and they showed comparable pH stability but considerably reduced thermal stability compared to the wt-YqjM. Next, we explored the operational window for the envisioned biotransformation. For cofactor (NADPH) regeneration we chose the well-known glucose dehydrogenase (GDH) system[45]. Using 10 mM of **1a** in aqueous media resulted in only 46% conversion. We attributed this

to the poor aqueous solubility of the reagents and therefore evaluated a range of water-miscible and –immiscible cosolvents. The presence of 5% v/v of water-miscible N,N-dimethylformamide (DMF), 1,2-dimethoxyethane (DME), dimethyl sulfoxide (DMSO) and acetone led to a decrease in conversion (Supplementary Table 7, entry 2-5), while in the presence of the water-immiscible isooctane (5 and 10% v/v) a significant improvement (up to 78%) was observed (Supplementary Table 7, entry 6-8). On the one hand this may be attributed to the negative influence of the water-soluble cosolvents on the stability of the biocatalyst (Supplementary Fig. 31). On the other hand, it may be assumed that the hydrophobic isooctane served as substrate reservoir and product sink for the likewise hydrophobic reagents and thereby contributed to alleviating the pronounced substrate inhibition of YqjM (Supplementary Fig. 32)[54,55]. For the bienzymatic reaction system a reaction temperature of 30 °C and a pH value of 7.5 were found to enable up to 96% conversion (Supplementary Table 7, entry 12). Attempts to further increase the substrate loading to 50 or 100 mM resulted in somewhat reduced conversions of 49% and 26%, respectively (Supplementary Table 7, entry 15 and 16), which was attributed to severe substrate inhibition even in the biphasic system. Notably, reducing the enzyme concentration to 6.0 μM in the presence of 50 μM NADP$^+$ still resulted in 91% conversion after 24 h, corresponding to a turnover number (TON) of 1517 for the enzyme (Supplementary Table 8).

**One-pot concurrent chemoenzymatic cascade**
Having established the enzymatic module, we turned our attention to constructing the chemoenzymatic cascade for the formal α-benzylation of cyclic carbonyl compounds. Following the method developed by Yin and co-workers[31], the water-stable organobismuth complex ([S(CH$_2$C$_6$H$_4$)$_2$Bi(OH$_2$)]$^+$[OSO$_2$C$_8$F$_{17}$]$^-$) was synthesized and used to catalyze the aldol condensation reaction between cyclic ketones and aryl aldehydes for the synthesis of exocyclic α,β-unsaturated ketones. n-Propylamine (n-PrNH$_2$) was added to activate ketone substrates by imine formation. To our delight, this catalyst exhibited high activity and selectivity under the reaction conditions predefined as suitable for the enzymatic reaction, providing the prerequisite for a concurrent cascade system. Indeed, following the reported conditions for the chemocatalytic step the desired product **1ap** was obtained in 92% (GC) yield and 98% ee: cyclopentanone (12 mM), benzaldehyde (10 mM), n-PrNH$_2$ (1.0 mM), organobismuth catalyst (0.1 mM), NADP$^+$ (50 μM) and glucose (20 mM), YqjM mutants (6.0 μM), and GDH (12 μM) in PBS (100 mM, pH 7.5) containing 10% v/v isooctane at 30 °C for 24 h.

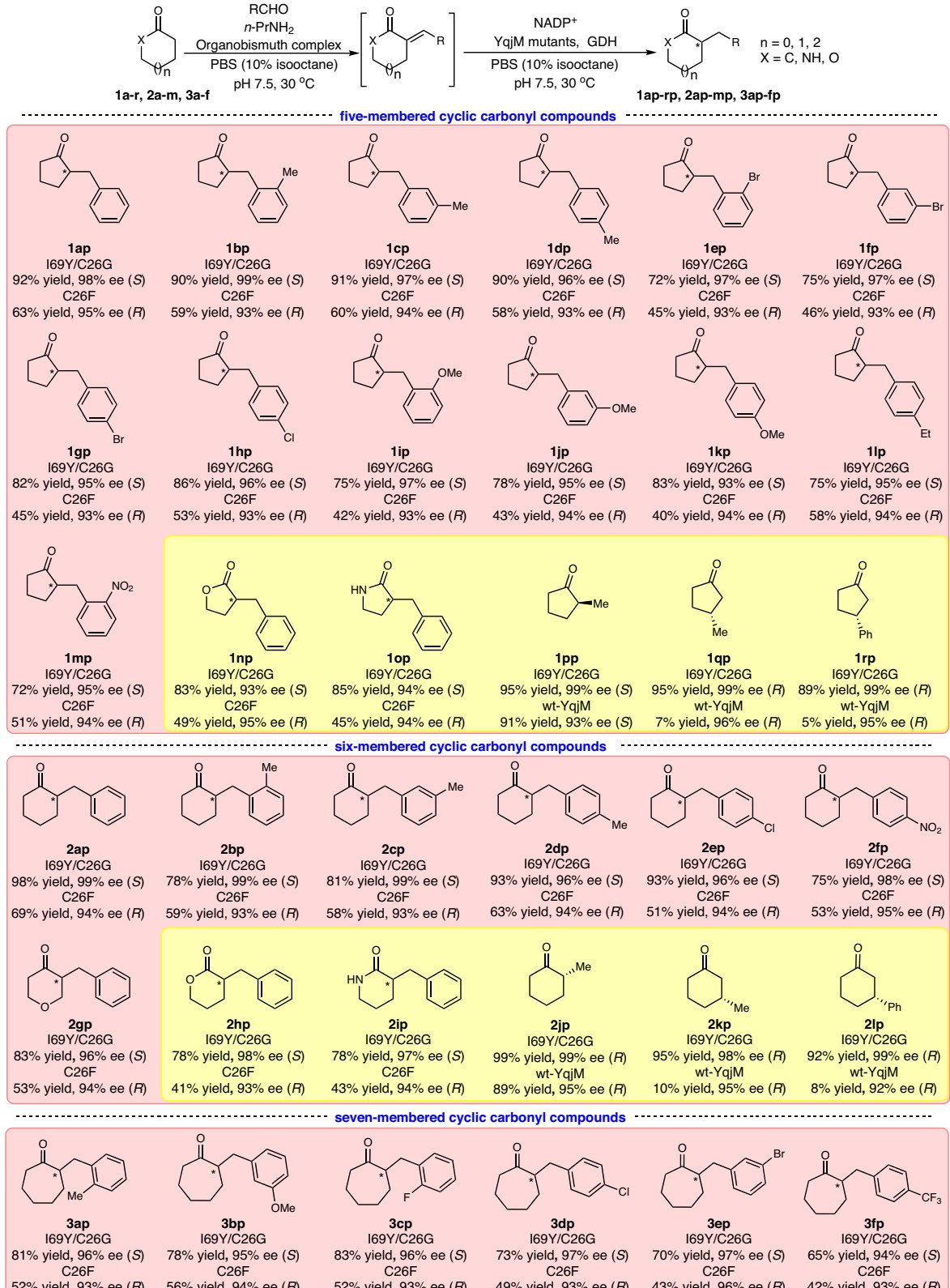

**Fig. 3 | Substrate scope study.** One-pot chemoenzymatic cascade for formal α-benzylation of cyclic ketones (pink shaded areas), and enzymatic reduction of various cyclic carbonyl compounds (yellow shaded areas).

Under the above-established conditions at 10 mM ketone concentration, the chemoenzymatic cascade achieved the direct conversion of simple (hetero)cyclic ketones to the corresponding chiral α-benzyl cyclic ketones, including five- (**1a-m**), six- (**2a-g**), and seven-membered (**3a-f**) cyclic ketones (Fig. 3). Notably, the nitro group (NO₂-), which often is also reduced in chemocatalytic reductions, stayed intact (**1m** and **2f**). The (*S*)-products were obtained by YqjM (I69Y/C26G) in 65-98% yields and 93-99% ee, while the (*R*)-products

**Fig. 4 | YqjM-based reaction scheme for the enantioselective synthesis of key chiral synthons. a** Enzymatic reduction of (+)-pulegone for the synthesis of (-)-menthone and (+)-isomenthone. **b** Multienzyme cascade for the stereocomplementary synthesis of 2-benzyl cyclohexanols. **c** Multienzyme cascade for the synthesis of the key intermediate of the active form of loxoprofen.

were obtained by YqjM (C26F) in 40-69% yields and 93-96% ee. It is worth mentioning here that the concurrent chemoenzymatic also enabled us to overcome the above-mentioned substrate inhibition issue. Starting from cyclopentanone (100 mM) and benzaldehyde (120 mM) under otherwise identical conditions, the desired **1ap** was obtained in approx. 90% yield and 97% ee. The time course of this reaction (Supplementary Fig. 33) reveals that the concentration of the YqjM substrate (**1a**) never exceeded 15 mM and thereby remained at around $v_{max}$ (Supplementary Fig. 32) of the enzymatic reduction reaction. Hence, we conclude that the pronounced substrate inhibition of YqjM by **1a** was also alleviated by its concurrent, in situ generation.

The two YqjM mutants also showed high activity towards α-arylidene lactones (**1n** and **2 h**) and lactams (**1o** and **2i**). However, our trials to generate the enone starting materials using the organobismuth catalyst were unsuccessful, which is why for these products the full cascade could not be established. It is also interesting to note that YqjM (I69Y/C26G), compared to the wt parent, exhibited higher activity towards 2-substituted endocyclic enones (**1p** and **2j**). Also the activity of this mutant towards 3-substituted endocyclic enones (**1q,r** and **2k,l**), which again poorly accepted by wt-YqjM was significantly improved. Hence, we are convinced that YqjM (I69Y/C26G), beyond the scope of this contribution is a valuable addition to the ene-reductase toolbox.

## YqjM-based reaction scheme for the synthesis of key chiral synthons
To further demonstrate the synthetic utility of the new, enantiocomplementary YqjM mutants, we investigated their application in enantioselective synthesis of some key chiral synthons. The reduction of (+)-pulegone, for example, a challenging substrate for currently available ERs[41], resulted in (-)-menthone and (+)-isomenthone in 53% yield (>99% ee) and 98% yield (>99% ee), respectively (Fig. 4a).

2-Benzyl cyclic alcohols, bearing two contiguous stereocenters, are frequent motifs in biologically active molecules and pharmaceutically active ingredients. Obtaining full control over the two

stereocenters, however remains challenging. We therefore set out to expand out stereocomplementary enone reduction with a stereoselective carbonyl reduction step (Fig. 4b). For this, we took advantage of the previously characterized NADH-dependent alcohol dehydrogenase from *Rhodococcus ruber* (ADH-A for an (S)-selective ketoreduction)[56] and the (R)-selective and NADPH-dependent *LK*-ADH[57] from *Lactobacillus kefir*. All four theoretically possible stereoisomers of 2-benzylcyclohexan-1-ol (**4**) were obtained in moderate to high yields (63-95%) with high enantioselectivity (94-99% ee) and excellent diastereoselectivity (>99:1 dr). By the YqjM&ADH-A cascade, the key intermediate of the active form of loxoprofen, (1S,2R)-**5**[58], was synthesized in 67% yield with 99% ee and 99:1 dr (Fig. 4c).

In conclusion, we have designed a pair of enantiocomplementary YqjM mutants for the stereodivergent reduction of exocyclic α,β-unsaturated carbonyl compounds including ketones, lactones, and lactams, thereby broadening the toolbox of ER catalysis for organic synthesis. The key to success was to optimize the hydride attack distance/angle and, more importantly, the generation of a new proton delivery cascade.

Combination to a chemocatalytic aldol condensation not only enables starting from simple ingredients but also allows to circumvent of inhibitory effects by in situ generation of the substrates.

The YqjM mutants alone or in combination with other catalysts achieved the synthesis of several key chiral synthons, further demonstrating their usefulness in synthetic chemistry. This work provides not only a new route to chiral α-benzyl cyclic carbonyls but also a potential roadmap for chemoenzymatic retrosynthetic analysis and cascade design.

## Methods
### Model generation and substrate docking
To analyze the possible effects of the different mutations on the YqjM activity, molecular docking studies between the wt-YqjM or mutants and their substrates were performed using Autodock 4.2 version with Lamarckian Genetic Algorithm (LGA). The structure of the wt-YqjM was obtained from PDB (PDB code: 1Z41), and the structures of the YqjM

mutants were generated by SWISS-MODEL using the structure of wt-YqjM as the template. All non-amino acid atoms but FMN have been deleted in order to obtain an open state active site. All the structures of the docked substrates were prepared using the software ChemBio-Draw 3D. Prior to docking, for each protein structure, all water molecules were removed, and the polar hydrogen atoms were added. According to the stereoselectivity of the wt-YqjM, the pro-(S) pose of the prochiral substrates was docked into the active site. The GridBox parameters for docking the substrates to the enzymes were determined (grid center coordinates: $x = -7.534$, $y = 15.364$, $z = 19.413$; size coordinates: $x = 40$, $y = 40$, $z = 40$). The docking studies were performed with 100 runs using LGA, and the lowest energy cluster obtained was further used to analyze binding affinity and modes.

### Determination of enzymatic kinetic parameters

All kinetic experiments were conducted using a SpectraMax190 (Molecular devices, America) in PBS (100 mM, pH 7.5) and following the oxidation of NADPH at λ of 340 nm ($\varepsilon = 6220 \, M^{-1} \, cm^{-1}$) at 30 °C for 10 min. The initial rates were obtained by measuring the absorbance change between time points, not just a single measurement at the 10-minute time point. For a standard activity assay, an assay reaction mixture (1 mL) contained 10 μM enzyme, a certain amount of substrate (0-15 mM), 1 mM NADPH, and 1 mL PBS (100 mM, pH 7.5). The initial rates were calculated from the linear range of the fitted trend line of the progress curve. Kinetic parameters ($k_{cat}$ and $K_M$) and respective standard errors were determined by measuring the activities at different substrate concentrations and fitting the activity versus substrate concentration data to the Michaelis-Menten equation using Origin 8.6 (Supplementary Figs. 9–22).

### General procedure for enzymatic reduction and one-pot concurrent chemoenzymatic cascade

A 25-mL round-bottomed flask was charged with YqjM variants (6.0 μM), GDH (12 μM), NADP$^+$ (50 μM), glucose (20 mM), and PBS (5 mL, 100 mM, pH 7.5) containing 10% v/v isooctane. For enzymatic reduction process, substrate (exocyclic α,β-unsaturated carbonyl compounds, 10 mM) was added to the above-mixed solution, whereas for one-pot concurrent chemoenzymatic cascade, cyclic ketones (12 mM), aryl aldehydes (10 mM), n-PrNH$_2$ (1.0 mM), and organo-bismuth catalyst (0.1 mM) were added to the above-mixed solution. Then, the flask was capped tightly, and the reaction mixture was stirred at 30 °C for 24 h. Upon completion, the reaction mixture was extracted with Et$_2$O (5 mL x 3), and the organic phase was washed with saturated brine, and dried over anhydrous Na$_2$SO$_4$. Volatiles were removed under a vacuum. The crude product was purified by column chromatography using petroleum ether and ethyl acetate (8/1, v/v) as eluent to afford the desired product. Enantiomeric excess was determined by GC with a CP-Chiralsil-DEX CB capillary column. Characterization of products can be found in the Supplementary Methods. Analytical reference compounds were synthesized as reported in Supplementary Methods.

### General procedure for multienzyme cascade for the synthesis of (1 S,2 R)−5

Exocyclic enone **1 g** (10 mM), NADP$^+$ (1 mM), NAD$^+$ (1 mM), glucose (20 mM), YqjM (C26F, 0.1 mM), GDH (0.2 mM), and ADH-A (0.5 mM) were added to a 25-mL round-bottomed flask charged with PBS (5.0 mL, 100 mM, pH 7.5) containing 10% v/v isooctane. The flask was capped tightly, and the reaction mixture was stirred at 30 °C for 48 h. Upon completion, the reaction mixture was extracted with Et$_2$O (5 mL x 3), and the organic phase was washed with saturated brine, and dried over anhydrous Na$_2$SO$_4$. Volatiles were removed under a vacuum. The crude product was purified by column chromatography using petroleum ether and ethyl acetate (4/1, v/v) as eluent. Enantiomeric excess was determined by HPLC with a Chiralpak AD-3 column.

Characterization of the product can be found in the Supplementary Methods. Analytical reference compound was synthesized as reported in Supplementary Methods.

### Reporting summary

Further information on research design is available in the Nature Portfolio Reporting Summary linked to this article.

## Data availability

The raw data underlying Table 1, Supplementary Figs. 9-33, and Supplementary Tables 4-8 are provided as a Source Data file. The other data that support the findings of this study are available within the manuscript, Supplementary Information, Source data file, or from the corresponding authors upon request. Source data are provided with this paper.

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

## Acknowledgements

This work was supported by the National Key Research and Development Program of China (No. 2023YFA0914500), the National Natural Science Foundation of China (Nos. 22378096, 22308083, 22178083 and 22078081), the Natural Science Foundation of Hebei Province (Nos. B2023202014 and B2022202014), the S&T program of Hebei (Nos. 21372805D and 21372804D), and the Natural Science Foundation of Tianjin City (No. 20JCYBJC00530).

## Author contributions

Y.J. and F.H. supervised the project. Y.L. and Y.J. conceived the idea. T.M., L.M., and Y.H. performed the protein design with technical help from L.Z., J.G., and J.B. Z.G. performed the DFT calculations. Z.G. and G.L. performed the kinetic analysis. Y.L. and T.M. participated in synthesizing and analyzing the compounds. Y.L., Y.B., Y.J., and F.H. co-wrote the paper.

## Competing interests

The authors declare no competing interests.
