## [Peer Review File · Nature Communications]

REVIEWER COMMENTS

Reviewer #1 (Remarks to the Author):

I appreciated to read and review this manuscript. The authors presented a very interesting, combined exp/theo study in which the stereo and chemoselectivity of an ER enzyme was redesigned for chemical synthesis purposes. The authors succeeded in "reprogramming" the YqjM catalysis via directed evolution by obtaining two mutants with i) high efficiency and ii) complementary enantioselectivity. The authors also showed that these mutants may operate in tandem with an organobismuth catalyst to afford the one pot synthesis of chemical scaffolds of relevance. The study is well-approached, well-structured, and the outcomes fulfilled the expectations.

The computational study, matching my scientific expertise, was an essential component of this investigation, and both docking and DFT results were optimally matched with the experimental outcomes.

The manuscript is also well-written and organized, although some aspects can be improved to make it more reader-friendly.

Please find below my comments and ideas to get some improvement to the manuscript content, as well as some questions (stimulated by my curiosity) for the authors.

Comments:

1) Molecular docking

The description of the docking methodology, reported in SI, lacks several information: which algorithm has been employed? How the structures of mutants were generated: did you use homology modelling? How did you assign the side chain conformation to the mutated residues? The reliability of docking results regarding the substrate binding at mutants is biased by the way the mutated structures are obtained. I think that including a 'Computational details' note in the SI may be helpful (see also a similar comments below at 2)).

- page 7, line 128, "Although difficult to explain" :

This sentence does not sound good to me, because its meaning is somewhat in the middle between “we know the cause of this effect but we are not able to deliver this information to the reader” and “in fact we have no idea of what is the cause of this effect”. Thus, my suggestion is either adding some more information if you have or just saying “Unexpectedly”.

- page 7, line 138, "edge-to-face pi-pi interaction" :

While in the face-to-face pairing of two aromatic rings the pi-pi interaction is reasonable, I am not sure that the orbital interaction in the edge-to-face pairing is pi-pi. The combination of empty sigma* orbitals on hydrogen atoms (edge) with the occupied pi orbitals of the other ring (face) should be, in my opinion, more reasonable. Could you provide references or data to substantiate the edge-to-face pi-pi interaction? Otherwise, my suggestion is just to delete “pi-pi” and leave "edge-to-face interaction".

2) DFT calculations

- page 8, line 154 "The stationary point energies ... reported method.48 " :

The way the authors cited ref 48 is not appropriate. Ref 48 is a multilayered computational study in which the core model of YqjM is obtained by a combination of MD/QM/MM calculations. Based on what is currently reported in SI, it is not clear how the authors obtained the geometry of the YqjM core model. In the note of Table S5, they specify that calculated energies are relative to the energy of the reactant complex obtained as Boltzmann-weighted average of "the energies calculated for the profiles of the substrate binding pose" at the B3LYP/6-31G(d)+ level of theory, while in the caption of Figure S6, the authors reported that they performed optimization and frequency calculations at the B3LYP/6-31G* level of theory.

The narrative of the computational details must be improved. I suggest to report a 'computational details' note in SI that may include either only DFT or both docking and DFT details. The DFT note must clearly state the source of input structures, the level of theory used for optimization/frequency and single point calculations

The source of input structures must be clarified because the relevance of the DFT study depends on it. Indeed, the mutual poses of Tyr residues bias the DFT investigation of the proton transfer pathways. Did the authors freeze or constrain the position of some Tyr atoms to maintain the poses assumed in the enzyme? This information must be specified. In case the DFT results were obtained without constraints, the authors have to clearly state the impact of DFT optimization on the mutual poses of Tyr residues compared to that assumed in the whole enzyme.

- page 8, line 165 "the Tyr28 in wt-YqjM has no proton delivery ability":

This is very interesting. Your finding highlights that handling the enzymatic selectivity is viable, and the variation of the “parterre” of the active residues can be accomplished to redesign the catalysis.

Just a question for my curiosity: Is the Cys26 thiol involved in the proton transfer pattern in the wt biocatalysis? The pKa values of Tyr OH and Cys-SH are not that different.

3) Establishment of the enzymatic reaction conditions

- page 11, line 224 "to the better biocompatibility of the hydrophobic isooctane" :

This is quite generic. Perhaps the authors meant that this enzyme may place at the liq-liq interface, but, in this case, this hypothesis would not be an alternative but just a corroboration to the second hypothesis, i.e. the enzyme picks the substrate at the liq-liq phase and operates its conversion in the aqueous phase.

4) One-pot concurrent chemoenzymatic cascade

- page 12, 255-257 "Apparently, the in situ ... concentration for YqjM.":

I would rephrase in 'Apparently, the in situ generation was effective to maintain the concentration of the exocyclic enone (1a) below its critical inhibitory concentration for YqjM.'

- page 12, line 256 "critical" : Which percentage of inhibition can be defined critical, 50% ? 90% ?

5) YqjM-based reaction scheme for the synthesis of key chiral synthons

- page 16, line 295 "staring" : Should it be "starting" instead?

- page 16, line 295/296 "starting materials": Could it be replaced by "ingredients" ?

- page 16, line 296/297 "starting materials": Could it be replaced by "substrates" ?

Reviewer #2 (Remarks to the Author):

The paper describes the investigation of a one-pot concurrent chemo-enzymatic cascade reaction for the α arylmethylation of cyclic ketones. The first step consists of an aldolic condensation catalysed by an organobismuth derivative. In the second step the unsaturated ketone is submitted to biocatalysed enantioselective reduction of the C=C double bond by using an ene-reductase. The ene-reductase

YqjM was suitably mutated through a mechanism-guided directed evolution approach, leading to the discovery of a pair of enantiocomplementary YqjM mutants for the reduction of exocyclic alpha,beta-unsaturated compounds.

I think that the paper doesn't meet the criteria of novelty and outstanding research results to justify the publication in Nature communications. Enzyme engineering is nowadays routinely performed to adapt enzyme to specific needs and publication on a more specialised journal, such as Advanced synthesis & catalysis, would be more appropriate.

The idea to combine an aldolic reaction to an ER-mediated in a cascade reaction is not new: see for example <http://dx.doi.org/10.1002/cctc.201501241> by Gröger et al.

DFT calculations have been done to suggest a mechanism for the protonation of the intermediate enolate. The key role played on C=C reduction by a Tyr residue in the active site of ene-reductases was discussed and assessed long ago by Kohli and Massey: 10.1074/jbc.273.49.32763.

The reduction of the C=C double bond of pulegone using the two enantiocomplementary mutants of YqjM cannot afford the two enantiomers of menthone, but it produces (+)-menthone in one case and (+)-isomenthone in the other one.

Chiral synthon 4 is shown to be interesting, because structurally related to the active form of loxofen, but the conversion of 4 into the active drug is not described. On the other hand, the reduction of the arylidene ketone, precursor of loxofen, is less likely to be successful.

Reviewer #3 (Remarks to the Author):

In this manuscript, Liu et al. reported the design of a one-pot chemoenzymatic process to synthesize chiral alpha-benzyl ketones by combining aldol condensation with conjugate reduction with an engineered ene reductase. The authors began by investigating the viability of the biocatalytic reaction on a model substrate with the ene reductase YqjM. This was followed by an engineering campaign that enlisted the help of molecular docking and DFT calculation to improve the activity of the enzyme. This campaign resulted in the identification of a double mutant with much improved specific activity and conversion. Based on their docking study, the authors hypothesized that restricting the size of the previously obtained active site pocket might lead to an enantioselectivity

switch. Thus, a single mutation C26F was shown to switch the product stereoconfiguration from S to R on their model substrate. After these enzymes were identified, the authors next investigated the best organic co-solvent for the reaction, identified a suitable NADPH regeneration method and then developed a chemoenzymatic process by introducing a Bi-catalyzed aldol reaction in the same pot. This process proved effective for the synthesis of a wide range of chiral branched ketones with moderate to high yields and high ee. The authors also demonstrated the utility of the process in the synthesis of two diastereomers of menthone and also in the synthesis of a chiral alcohol subunit of loxoprofen. Overall, I am supportive of accepting this manuscript for publication in Nature Communications. The authors have successfully engineered a useful variant of YqjM that can process a wide range of substrates with high conversion and ee and I anticipate that this variant will attract a lot of interest from the biocatalysis community. However, there are several issues that need to be addressed before the manuscript can be accepted, as found below.

The authors refer to the constructed motif as “alpha-arylmethyl”. I personally think that the term “alpha-benzyl” is better. See JACS 2010, 132, 13600 for the use of this term previously.

Line 87–88: “...highly challenging, especially no precedent to realize...” this feels like a run-on sentence and the last fragment after the comma is not a complete sentence. Consider breaking this up into two sentences.

Line 144: “relais” I am not aware of the use of this term before. Perhaps the sentence should be revised to “the importance of Tyr169 in proton relay.”?

Line 172-173: this should be revised to “the obtained mutant... has drastically-lowered activity...”

Line 225: “positive effects of the 2-liquid-phase-system” – can the authors be more specific here? Is there any relevant literature to be cited here?

Table 4a – (+) and (-) label usually refers to enantiomers but in this case, the 2 menthone isomers produced are diastereomers

Line 278: replace “frequent themes” with “frequent motifs”

Supporting Information:

- There are random “2”s everywhere on the left border of the SI. Please fix.
- While the authors have provided the sequence for YqjM, they did not provide any information on how the final plasmid was obtained. Please add this information.
- Michaelis-Menten plot: Please provide more details on how the plot was obtained. For example, what are the actual substrate concentrations that were used in determining the initial rates (instead of just listing the range of 0-8 mM)? Are the initial rates obtained from just a single measurement at 10 min time point or are there in between time points? What program was used to fit the Michaelis-Menten plot into the obtained data? It is also strange to me that the Michaelis-Menten plots were presented as just the final fitted curves with no data points and error bars presented. These should be added, along with the final k_{cat} and K_m for each plot as well as the R^2 of the fit.

Reviewer 1

I appreciated to read and review this manuscript. The authors presented a very interesting, combined exp/theo study in which the stereo and chemoselectivity of an ER enzyme was redesigned for chemical synthesis purposes. The authors succeeded in "reprogramming" the YqjM catalysis via directed evolution by obtaining two mutants with i) high efficiency and ii) complementary enantioselectivity. The authors also showed that these mutants may operate in tandem with an organobismuth catalyst to afford the one pot synthesis of chemical scaffolds of relevance. The study is well-approached, well-structured, and the outcomes fulfilled the expectations.

The computational study, matching my scientific expertise, was an essential component of this investigation, and both docking and DFT results were optimally matched with the experimental outcomes.

The manuscript is also well-written and organized, although some aspects can be improved to make it more reader-friendly.

Response: We thank the reviewer for these positive comments on the manuscript. We have done our best to improve the manuscript in her/his spirit.

Please find below my comments and ideas to get some improvement to the manuscript content, as well as some questions (stimulated by my curiosity) for the authors.

Comments:

(1) Molecular docking

The description of the docking methodology, reported in SI, lacks several information: which algorithm has been employed? How the structures of mutants were generated: did you use homology modelling? How did you assign the side chain conformation to the mutated residues? The reliability of docking results regarding the substrate binding at mutants is biased by the way the mutated structures are obtained. I think that including a 'Computational details' note in the SI may be helpful (see also a similar comments below at 2)).

Response: Following the reviewers advise we have added a section 'Computational details' to the SI to describe the methods used for model generation and substrate docking.

Modification: In the revised supporting information, page 5 we added:

Model generation and substrate docking

To analyze the possible effects of the different mutations on the YqjM activity, molecular docking studies between the wild-type enzyme or mutants and their substrates were performed using Autodock 4.2 version with Lamarckian Genetic Algorithm (LGA). The structure of the wild-type YqjM was obtained from PDB (PDB code: 1Z41), and the structures of the YqjM mutants were generated by SWISS-

MODEL using the structure of wild-type YqjM as the template. All non-amino acid atoms but FMN have been deleted in order to obtain an open state active site. All the structures of the docked substrates were prepared using the software ChemBioDraw 3D. Prior to docking, for each protein structure, all water molecules were removed, and the polar hydrogen atoms were added. According to the stereoselectivity of the wild-type YqjM, the *pro-(S)* pose of the prochiral substrates was docked into the active site. The GridBox parameters for docking the substrates to the enzymes were determined (grid center coordinates: $x = -7.534$, $y = 15.364$, $z = 19.413$; size coordinates: $x = 40$, $y = 40$, $z = 40$). The docking studies were performed with 100 runs using LGA, and the lowest energy cluster obtained was further used to analyze binding affinity and modes.”

-page 7, line 128, "Although difficult to explain":

This sentence does not sound good to me, because its meaning is somewhat in the middle between “we know the cause of this effect but we are not able to deliver this information to the reader” and “in fact we have no idea of what is the cause of this effect”. Thus, my suggestion is either adding some more information if you have or just saying “Unexpectedly”.

Response: We absolutely agree with the reviewer and have changed the manuscript text accordingly.

Modification: In the revised manuscript, page 7, line 127: “Although difficult to explain, the sulfur containing cysteine and methionine also increased the binding energy.” **was changed to** “Unexpectedly, the sulfur containing cysteine and methionine also increased the binding energy.”.

page 7, line 138, "edge-to-face pi-pi interaction":

While in the face-to-face pairing of two aromatic rings the pi-pi interaction is reasonable, I am not sure that the orbital interaction in the edge-to-face pairing is pi-pi. The combination of empty sigma* orbitals on hydrogen atoms (edge) with the occupied pi orbitals of the other ring (face) should be, in my opinion, more reasonable. Could you provide references or data to substantiate the edge-to-face pi-pi interaction? Otherwise, my suggestion is just to delete “pi-pi” and leave "edge-to-face interaction".

Response: We thank the reviewer for her/his comment. In fact edge-to-face pi-pi interactions have been described in the literature. For example in the references below, which we have now referenced in the revised manuscript.

(1) *Angew. Chem. Int. Ed.* **2017**, *56*, 5517-5521 (Figure 2a): “Adjacent residue Tyr97 makes an edge-to-face pi-pi interaction with a pClx₆ phenyl ring (C^ε...centroid 3.8 Å).”

(2) *J. Med. Chem.* **2018**, *61*, 4978-4992 (Figure 5b): “In this structure, Tyr64 adopts a slightly altered conformation in which its phenol ring is twisted to engage in an edge-to-face pi-pi interaction with the phenyl group of the inhibitor, forming a lid over the second pocket.”

(3) *J. Med. Chem.* **2022**, *65*, 8828-8842 (Figure 5a): “The main difference is the orientation of Phe85, which in 7AMG has flipped to make space for the long tail of compound **1**. In 5HI5 this residue instead caps the binding site and forms an edge-to-face pi-pi interaction with one of the aromatic rings of Pfizer **3**.”

Modification: In the revised manuscript, page 7, line 140: “Such interactions are well-documented in the literature.^{49,50}” was added.

Modification: In the revised manuscript, page 21, line 413: “

49. Heightman, T. D. et al. Fragment-based discovery of a potent, orally bioavailable inhibitor that modulates the phosphorylation and catalytic activity of ERK1/2. *J. Med. Chem.* **61**, 4978–4992 (2018).

50. Rennie, M. L., Doolan, A. M., Raston, C. L & Crowley P. B. Protein dimerization on a phosphonated calix[6]arene disc. *Angew. Chem. Int. Ed.* **56**, 5517-5521 (2017).” **was added.**

(2) DFT calculations

- page 8, line 154 "The stationary point energies ... reported method.48":

The way the authors cited ref 48 is not appropriate. Ref 48 is a multilayered computational study in which the core model of YqjM is obtained by a combination of MD/QM/MM calculations. Based on what is currently reported in SI, it is not clear how the authors obtained the geometry of the YqjM core model. In the note of Table S5, they specify that calculated energies are relative to the energy of the reactant complex obtained as Boltzmann-weighted average of "the energies calculated for the profiles of the substrate binding pose" at the B3LYP/6-31G(d)+ level of theory, while in the caption of Figure S6, the authors reported that they performed optimization and frequency calculations at the B3LYP/6-31G* level of theory.

The narrative of the computational details must be improved. I suggest to report a 'computational details' note in SI that may include either only DFT or both docking and DFT details. The DFT note must clearly state the source of input structures, the level of theory used for optimization/frequency and single point calculations

The source of input structures must be clarified because the relevance of the DFT study depends on it. Indeed, the mutual poses of Tyr residues bias the DFT investigation of the proton transfer pathways. Did the authors freeze or constrain the position of some Tyr atoms to maintain the poses assumed in the enzyme? This information must be specified. In case the DFT results were obtained without constraints, the authors have to clearly state the impact of DFT optimization on the mutual poses of Tyr residues compared to that assumed in the whole enzyme.

Response: The reviewer is absolutely right and we have added a ‘computational details’ for DFT calculations in the supporting information together with some appropriate references. In addition, the note of Table S5 and the caption of Figure S6 have been corrected.

Modification: In the revised supporting information, Page 8, “

DFT calculations

The DFT region was extracted from the YqjM (wild-type) structure obtained from PDB (PDB code: 1Z41) and the YqjM mutant structure generated by SWISS-MODEL, consisting of the substrate, the flavin mononucleotide (FMN) truncated at the C1 position, and the side-chain atoms of Y169 or Y69. Additionally, the water molecule located between the substrate and Y69 was included in the DFT region when considering water as a proton donor. All DFT calculations were carried out using the Gaussian 09 program.¹ The geometry optimizations of intermediates and transition states were performed using the B3LYP+D3/6-31G(d) level.² The SMD³ solvation model using the experimentally used solvent, water, was applied to estimate the effect of enzyme-surrounding by calculating single-point implicit solvation using a dielectric

constant of $\epsilon = 4$. Vibrational frequency calculations were performed for all stationary points to confirm if each optimized structure is a local minimum or a transition state structure. All optimized transition state structures have only one imaginary (negative) frequency, and all minima have no imaginary frequencies. To get more accurate energies, the single-point energy was calculated using the larger basis set 6-311++G(2d,p),⁴ denoted as the B3LYP+D3/6-311++G(2d,p) (SMD, solvent = water)//B3LYP+D3/6-31G(d) (SMD, solvent = water) level.⁵ It should be emphasized that although the Tyr residues were not constrained, the DFT optimization led to only negligible change in the mutual poses of the Tyr residues compared to those assumed in the enzyme, reflecting the reasonable real pocket environment.

1. Frisch, M. J. et al. Gaussian 09, revision A.1; Gaussian, Inc.: Wallingford, CT, 2009.
2. Lonsdale, R. & Reetz, M. T. Reduction of α , β -unsaturated ketones by old yellow enzymes: mechanistic insights from quantum mechanics/molecular mechanics calculations. *J. Am. Chem. Soc.* **137**, 14733-14742 (2015).
3. Marenich, A. V., Cramer, C. J. & Truhlar, D. G. Universal solvation model based on solute electron density and on a continuum model of the solvent defined by the bulk dielectric constant and atomic surface tensions. *J. Phys. Chem. B* **113**, 6378-6396 (2009).
4. Becke, A. D. Density-functional exchange-energy approximation with correct asymptotic behavior, *Phys. Rev.* **38**, 3098-3100 (1988).
5. Qu, G. et al. Computational insights into the catalytic mechanism of bacterial carboxylic acid reductase. *J. Chem. Inf. Model.* **59**, 832-841 (2019).

” was added.

Modification: In the revised manuscript, page 21, line 423: “

53. Qu, G. et al. Computational insights into the catalytic mechanism of bacterial carboxylic acid reductase. *J. Chem. Inf. Model.* **59**, 832-841 (2019).

” was added.

-page 8, line 165 "the Tyr28 in wt-YqjM has no proton delivery ability":

This is very interesting. Your finding highlights that handling the enzymatic selectivity is viable, and the variation of the “parterre” of the active residues can be accomplished to redesign the catalysis.

Just a question for my curiosity: Is the Cys26 thiol involved in the proton transfer pattern in the wt biocatalysis? The pKa values of Tyr OH and Cys-SH are not that different.

Response: We thank the reviewer for this interesting suggestion! A careful re-investigation of our experimental and computational experiments, however, yielded no indications pointing towards an involvement of Cys26. Considering the negligible

activity of Y169F, it can also be concluded that Cys26 in wt-YqjM is not involved as catalytic acid. Most likely, this can be attributed to the long distance between the Cys-SH and the alkene C α .

Modification: In the revised manuscript, Page 8, line 168: “The long distance between the Cys-SH and the alkene C α renders an involvement of Cys26 as proton source in the catalytic mechanism unlikely.” **was added.**

(3) Establishment of the enzymatic reaction conditions

- page 11, line 224 "to the better biocompatibility of the hydrophobic isooctane":

This is quite generic. Perhaps the authors meant that this enzyme may place at the liq-liq interface, but, in this case, this hypothesis would not be an alternative but just a corroboration to the second hypothesis, *i.e.* the enzyme picks the substrate at the liq-liq phase and operates its conversion in the aqueous phase.

Response: The reviewer is absolutely right noting that the original statement was too general. As shown in Table S7, the water-soluble co-solvents lead to a significant decrease in product formation, which we could attribute to the negative effects of these solvents on the stability of the biocatalyst (Figure S34). At the same time hydrophobic isooctane did not influence the enzyme activity to a similar extend as the water-miscible cosolvents (Figure S34). An in-depth-explanation of this observation would go beyond the scope of this contribution but similar effects have been observed previously. (e.g. C. Laane, et al Rules for Optimization of Biocatalysis in Organic Solvents, Biotechnol. Bioeng. 1987, 30, 81-87). Yet another (non-stability related) influence of the hydrophobic organic phase may have been that it served as substrate reservoir and product sink thereby alleviating the pronounced substrate/product inhibition issue of YqjM (Figure S35). (*ACS Catal.* 2014, 4, 4021-4026; *ACS Catal.* 2017, 7, 1295 –1300)

Modification: In the revised manuscript, Page 11, line 226: “We attribute these findings to the better biocompatibility of the hydrophobic isooctane (Figure S34) and possibly to some positive effects of the 2-liquid-phase-system formed.” **was changed to** “On the one hand this may be attributed to the negative influence of the water-soluble cosolvents on the stability of the biocatalyst (Figure S34). On the other hand, it may be assumed that the hydrophobic isooctane served as substrate reservoir and product sink for the likewise hydrophobic reagents and thereby contributed to alleviating the pronounced substrate inhibition of YqjM (Figure S35).”.

Modification: In the revised manuscript, page 21, line 425: “

54. Au, S. K., Bommarius, B. R., & Bommarius, A. S. Biphasic reaction system allows for conversion of hydrophobic substrates by amine dehydrogenases. *ACS Catal.* 4,

4021-4026 (2014).

55. Scalacci, N. et al. Unveiling the biocatalytic aromatizing activity of monoamine oxidases MAO-N and 6-HDNO: development of chemoenzymatic cascades for the synthesis of pyrroles. *ACS Catal.* **7**, 1295-1300 (2017).” **was added.**

(4) One-pot concurrent chemoenzymatic cascade

- page 12, 255-257 "Apparently, the in situ ... concentration for YqjM.":

I would rephrase in 'Apparently, the *in situ* generation was effective to maintain the concentration of the exocyclic enone (**1a**) below its critical inhibitory concentration for YqjM."

- page 12, line 256 "critical": Which percentage of inhibition can be defined critical, 50% ? 90% ?

Response: The reviewer is absolutely right noting that our initial phrasing was too vague. Following the reviewer’s comment we performed additional experiments determining the in situ concentration of the substrate **1a**. These experiments showed that the concentration of **1a** did not reach inhibitory values (please also see Figures S35 and S36).

Modification: In the revised manuscript, Page 11, line 259: “The time course of this reaction (Figure S36) reveals that the concentration of the YqjM substrate (**1a**) never exceeded 15 mM and thereby enabled the YqjM-catalyzed reduction to run around v_{max} (Figure S35). Hence, we conclude that the pronounced substrate inhibition of YqjM by **1a** was also alleviated by its concurrent, *in situ* generation.”.

Modification: In the revised supporting information, Page 50, “

Figure S36 Reaction time curves of chemoenzymatic cascade for α -benzylation of cyclic ketones.” **was added.**

(5) YqjM-based reaction scheme for the synthesis of key chiral synthons

- page 16, line 295 "starting": Should it be "starting" instead?

- page 16, line 295/296 "starting materials": Could it be replaced by "ingredients"?

- page 16, line 296/297 "starting materials": Could it be replaced by "substrates"?

Response: Corrected as suggested by the reviewer.

Modification: In the revised manuscript, Page 16, line 298: “Combination to a chemocatalytic aldol condensation not only enable staring from simple starting materials but also allow to circumvent inhibitory effects by *in situ* generation of the starting materials.” **was changed to** “Combination to a chemocatalytic aldol condensation not only enable starting from simple ingredients but also allow to circumvent inhibitory effects by *in situ* generation of the substrates.”

Responses to reviewer 2's comments

The paper describes the investigation of a one-pot concurrent chemo-enzymatic cascade reaction for the α -arylmethylation of cyclic ketones. The first step consists of an aldolic condensation catalysed by an organobismuth derivative. In the second step the unsaturated ketone is submitted to biocatalysed enantioselective reduction of the C=C double bond by using an ene-reductase. The ene-reductase YqjM was suitably mutated through a mechanism-guided directed evolution approach, leading to the discovery of a pair of enantiocomplementary YqjM mutants for the reduction of exocyclic α,β -unsaturated compounds.

(1) I think that the paper doesn't meet the criteria of novelty and outstanding research results to justify the publication in Nature communications. Enzyme engineering is nowadays routinely performed to adapt enzyme to specific needs and publication on a more specialised journal, such as Advanced synthesis & catalysis, would be more appropriate.

Response: We thank the reviewer for her/his honest evaluation of the manuscript. Of course the reviewer is right noting that enzyme engineering nowadays is a technique of the state-of-the-art. We would, however like to point out that the expansion of the substrate scope reported here is unprecedented. Compared to engineering efforts on ERs such as reported by Reetz and coworkers (e.g. 10.1002/adsc.200900644) mainly targeted expanding the scope and selectivity towards β -substituted starting materials whereas α -substituted enones, especially with larger substituents, are 'notoriously difficult substrates' (e.g. 10.1021/acscatal.8b00624; 10.1016/j.cbpa.2014.01.019; 10.1016/j.cbpa.2007.02.025). From this point-of-view, the results presented here will certainly encourage others to engineer ERs for enlarged substrate acceptance and increased synthetic usefulness and/or just use the mutants we have provided.

Therefore, we believe that this study does not represent 'yet another enzyme engineering study' and deserved publication in a prime journal as *Nature Communications*.

(2) The idea to combine an aldolic reaction to an ER-mediated in a cascade reaction is not new: see for example <http://dx.doi.org/10.1002/cctc.201501241> by Gröger et al.

Response: We apologize for having omitted this previous work in our original manuscript. Of course the reviewer is right noting that Gröger and coworkers have previously reported the ER-catalyzed reduction of enals that had been obtained in a separate previous aldol condensation step. The appropriate reference has now been included in the revised manuscript. We, would, however, like to draw the reviewer's attention to the fact that in this work Gröger and coworkers used a sequence of reaction

steps (including product isolation in between the different reaction steps) rather than a ‘one-pot-one-step’ cascade such as reported here. We believe that this represents a conceptual difference of our work with the previous report by Gröger and coworkers.

Modification: In the revised manuscript, Page 4, line 68: “The group around Gröger combined an organocatalytic aldol condensation reaction with an enzymatic (yet non-stereoselective) C=C bond reduction.²⁹ The incompatibility of the reagents and reaction conditions, however, forced the authors to perform the reaction in individual steps including intermediate product isolation.” **was added.**

(3) DFT calculations have been done to suggest a mechanism for the protonation of the intermediate enolate. The key role played on C=C reduction by a Tyr residue in the active site of ene-reductases was discussed and assessed long ago by Kohli and Massey: 10.1074/jbc.273.49.32763.

Response: We thank the reviewer for drawing our attention to the fact that even before Pietruszka, Massey had discussed the catalytic role of the Tyr169 in other ER homologues. We have added this reference and have modified the manuscript text to avoid a possible interpretation that we claim this finding. We would, however, like to draw the reviewer’s attention to the fact that the possible involvement of Tyr69 in the catalytic mechanism has not been discussed before.

Modification: The corresponding passage (Page 7, line 145) in the revised manuscript now reads: “Kohli and Massey as well as Pietruszka and coworkers previously emphasized the importance of the catalytic Tyr as proton relay.⁴⁵ Therefore, the mutants Y169F and I69Y/Y169F were created. While Y169F exhibited almost no catalytic activity, I69Y/Y169F still displayed a moderate activity (53.5 U/g_{protein}). Thus, the newly introduced Tyr69 not only accommodated the substrate binding in a favorable mode, but also participated in proton delivery during catalysis.”

(4) The reduction of the C=C double bond of pulegone using the two enantiocomplementary mutants of YqjM cannot afford the two enantiomers of menthone, but it produces (+)-menthone in one case and (+)-isomenthone in the other one.

Response: We apologize for this mistake and thank the reviewer for drawing our attention to it. The erroneous nomenclature was corrected.

(5) Chiral synthon 4 is shown to be interesting, because structurally related to the active form of loxofen, but the conversion of 4 into the active drug is not described. On the other hand, the reduction of the arylidene ketone, precursor of loxofen, is less likely to be successful.

Response: The reviewer is absolutely right noting that the chiral synthon 4 (in Table 4) was not likely a precursor to loxoprofen. To address this question, as shown below, the key intermediate ((1*S*,2*R*)-5) of the active form of loxoprofen reported by Zhou group (*J. Am. Chem. Soc.* 2010, 132, 4538-4539) have been synthesized by the YqjM&ADH-A cascade.

J. Am. Chem. Soc. 2010, 132, 4538-4539 (Scheme 3):

The revised Table 4:

Modification: In the revised manuscript, page 16, line 290: “By this way, the (1*S*,2*R*)- α -benzyl cyclopentanol (4), the key fragment of the active form of loxoprofen,⁷ was synthesized in 70% yield with 99% ee and 99:1 dr.” **was changed to** “By the YqjM&ADH-A cascade, the key intermediate of the active form of loxoprofen, (1*S*,2*R*)-5,⁵⁸ was synthesized in 67% yield with 99% ee and 99:1 dr.”.

Modification: In the revised manuscript, page 22, line 435: “

58. Xie, J. B. et al. Highly enantioselective hydrogenation of α -arylmethylene cycloalkanones catalyzed by iridium complexes of chiral spiro aminophosphine ligands. *J. Am. Chem. Soc.* 132, 4538-4539 (2010).” **was added.**

Modification: In the revised supporting information, page 16:“

The synthesis of *racemic-trans*-5

The reduction of C=O double bond according to the procedure reported by Lautens:¹⁷ A 50 mL round-bottomed flask equipped with a magnetic stir bar was charged with the exocyclic enone **1g** (5 mmol) and methanol (10 mL). At $0\text{ }^\circ\text{C}$, NaBH_4 (5 mmol) was added portion-wise. The mixture was stirred for 3 hours at room temperature or until full conversion is observed via TLC monitoring. The solution was then concentrated, redissolved in ethyl acetate or dichloromethane (15 mL), washed with water (10 mL), and brine (10 mL). The organic phase was then dried over MgSO_4 , filtered, concentrated *in vacuo*, and carried on to the next step.

The reduction of C=C double bond according to the procedure reported by Lautens:¹⁸ A solution of the allylic alcohol intermediate obtained in the first step in 10 mL Et_2O was added slowly to a 50 mL dry three-neck flask containing LiAlH_4 (10 mmol) and 10 mL dry Et_2O under N_2 atmosphere at $0\text{ }^\circ\text{C}$. The mixture was stirred at room temperature for 0.5 h, and then was refluxed for 5 h. After cooling to $0\text{ }^\circ\text{C}$ the reaction was quenched with small amount of water, the reaction mixture was treated with 1N HCl (30 mL) and extracted with ethyl acetate ($3 \times 30\text{ mL}$). The combined organic layer was washed with water, saturated NaHCO_3 solution in sequence, and dried over anhydrous Na_2SO_4 . The solvent was removed in vacuum and the residual was purified by silica gel column chromatography (EtOAc /petroleum ether 1:4) to yield *racemic-trans*-5.

Multienzyme cascade for the synthesis of (1*S*,2*R*)-5

Exocyclic enone **1g** (10 mM), NADP^+ (1 mM), NAD^+ (1 mM) and glucose (20 mM), YqjM (C26F, 0.1 mM) and GDH (0.2 mM), ADH-A (0.5 mM) and PBS (5.0 mL, 100

mM, pH 7.5) containing 10% v/v isooctane were added to a 25-mL round-bottomed flask. The mixture was stirred at 30 °C for 48 h. After the reaction was completed, the reaction solution was extracted with Et₂O (5 mL x 3), and the organic phase was dried using anhydrous Na₂SO₄. The solvent was concentrated in vacuo to obtain the crude products. The products were purified by column chromatography using petroleum ether and ethyl acetate (4/1, v/v) as eluent. The values of ee were determined by HPLC (Chiralpak AD-3 column, UV detection at 210 nm, eluent: *n*-hexane/2-propanol =80:20, flow 1 mL/min). ¹H NMR (400 MHz, CDCl₃) δ 7.40 (d, *J* = 8.3 Hz, 2H), 7.07 (d, *J* = 8.3 Hz, 2H), 3.88 (dd, *J* = 11.9, 5.8 Hz, 1H), 2.75 (dd, *J* = 13.6, 6.5 Hz, 1H), 2.45 (dd, *J* = 13.6, 8.7 Hz, 1H), 1.96 (tt, *J* = 12.7, 6.7 Hz, 2H), 1.85 – 1.75 (m, 1H), 1.74 – 1.66 (m, 1H), 1.63 – 1.53 (m, 2H), 1.40 (s, 1H), 1.24 (dd, *J* = 8.5, 5.2 Hz, 1H). ¹³C NMR (100 MHz, CDCl₃) δ 140.31, 131.64, 130.80, 119.89, 78.59, 49.87, 39.30, 34.51, 29.82, 21.65.” was added.

HPLC (revised supporting information, page 183):

Signal: DAD1C,Sig=210,4 Ref=360,100

RT [min]	Type	Width [min]	Area	Height	Area%
5.231	BV	0.33	4877.76	1021.34	99.26
5.499	VV	0.24	36.22	4.27	0.74
		Sum	4913.97		

NMR (revised supporting information, page 145):

Responses to reviewer 3's comments

In this manuscript, Liu et al. reported the design of a one-pot chemoenzymatic process to synthesize chiral alpha-benzyl ketones by combining aldol condensation with conjugate reduction with an engineered ene reductase. The authors began by investigating the viability of the biocatalytic reaction on a model substrate with the ene reductase YqjM. This was followed by an engineering campaign that enlisted the help of molecular docking and DFT calculation to improve the activity of the enzyme. This campaign resulted in the identification of a double mutant with much improved specific activity and conversion. Based on their docking study, the authors hypothesized that restricting the size of the previously obtained active site pocket might lead to an enantioselectivity switch. Thus, a single mutation C26F was shown to switch the product stereoconfiguration from S to R on their model substrate. After these enzymes were identified, the authors next investigated the best organic co-solvent for the reaction, identified a suitable NADPH regeneration method and then developed a chemoenzymatic process by introducing a Bi-catalyzed aldol reaction in the same pot. This process proved effective for the synthesis of a wide range of chiral branched ketones with moderate to high yields and high ee. The authors also demonstrated the utility of the process in the synthesis of two diastereomers of menthone and also in the synthesis of a chiral alcohol subunit of loxoprofen. Overall, I am supportive of accepting this manuscript for publication in Nature Communications. The authors have successfully engineered a useful variant of YqjM that can process a wide range of substrates with high conversion and ee and I anticipate that this variant will attract a lot of interest from the biocatalysis community. However, there are several issues that need to be addressed before the manuscript can be accepted, as found below. Overall the paper has potential but several things are unclear and must be addressed:

Response: We thank the reviewer for sharing our enthusiasm on the proposed reaction scheme! We have done our best to improve the quality of the manuscript in her/his spirit.

(1) The authors refer to the constructed motif as “alpha-arylmethyl”. I personally think that the term “alpha-benzyl” is better. See JACS 2010, 132, 13600 for the use of this term previously.

Response: Corrected throughout the manuscript as suggested by the reviewer.

(2) Line 87–88: “...highly challenging, especially no precedent to realize...” this feels like a run-on sentence and the last fragment after the comma is not a complete sentence. Consider breaking this up into two sentences.

Response: Following the reviewer's suggestion we have improved the clarity of the

original statement.

Modification: In the revised manuscript, page 5, line 87: “The substrate scope of ERs is generally limited to acyclic and endocyclic compounds, with exocyclic substrates remaining highly challenging, especially no precedent to realize the enzymatic reduction of α -arylidene cyclic carbonyl compounds.” **was changed to** “The substrate scope of ERs is generally limited to acyclic and endocyclic compounds, with exocyclic substrates remaining highly challenging.³⁸⁻⁴⁰ Particularly, to the best of our knowledge, there is currently no enzymatic reduction of α -arylidene cyclic carbonyl compounds known.”.

(3) Line 144: “relais” I am not aware of the use of this term before. Perhaps the sentence should be revised to “the importance of Tyr169 in proton relay.”?

Response: Changed as suggested by the reviewer.

Modification: In the revised manuscript, page 5, line 143: “...the importance of Tyr169 as proton relais” **was changed to** “...the importance of the catalytic Tyr as proton relay.”.

(4) Line 172-173: this should be revised to “the obtained mutant... has drastically-lowered activity...”

Response: Changed as suggested by the reviewer.

Modification: In the revised manuscript, line 172: “the obtained mutant I69Y/C26G/Y28F drastically lowered the activity...” **was changed to** “the obtained mutant I69Y/C26G/Y28F has drastically-lowered activity...”.

(5) Line 225: “positive effects of the 2-liquid-phase-system” –can the authors be more specific here? Is there any relevant literature to be cited here?

Response: We fully agree that the original statement was indeed not clear enough. Please also see our reply to reviewer#1

In the revised manuscript, this was changed to: On the one hand this may be attributed to the negative influence of the water-soluble cosolvents on the stability of the biocatalyst (Figure S34). On the other hand, it may be assumed that the hydrophobic isoctane served as substrate reservoir and product sink for the likewise hydrophobic reagents and thereby contributed to alleviating the pronounced substrate inhibition of YqjM (Figure S35).

Also some representative literature examples have been added:

Modification: In the revised manuscript, page 21, line 425: “

54. Au, S. K., Bommarius, B. R., & Bommarius, A. S. Biphasic reaction system allows for conversion of hydrophobic substrates by amine dehydrogenases. *ACS Catal.* **4**, 4021-4026 (2014).

55. Scalacci, N. et al. Unveiling the biocatalytic aromatizing activity of monoamine oxidases MAO-N and 6-HDNO: development of chemoenzymatic cascades for the synthesis of pyrroles. *ACS Catal.* **7**, 1295-1300 (2017).” **was added.**

(6) Table 4a (+) and (-) label usually refers to enantiomers but in this case, the 2 menthone isomers produced are diastereomers.

Response: We apologize for this mistake and thank the reviewer for drawing our attention to it. The erroneous nomenclature was corrected.

(7) Line 278: replace “frequent themes” with “frequent motifs”.

Response: Changed as suggested by the reviewer.

Modification: In the revised manuscript, page 15, line 282: “ α -Arylmethylene cyclohexanols, bearing two contiguous stereocenters, are frequent themes in biologically active molecules and in pharmaceutically active ingredients.” was changed to “ α -Benzylene cyclohexanols, bearing two contiguous stereocenters, are frequent motifs in biologically active molecules and in pharmaceutically active ingredients.”

(8) There are random “2”s everywhere on the left border of the SI. Please fix.

Response: Changed as suggested by the reviewer.

(9) While the authors have provided the sequence for YqjM, they did not provide any information on how the final plasmid was obtained. Please add this information.

Response: Added as suggested by the reviewer.

Modification: In the revised supporting information, Page 50, “

Construction of plasmid: Plasmid pET28b(+) (Novagen, Germany) was used as

the expression vector for the wild-type YqjM and the mutants constructed in this study. The gene YqjM encoding for the wild-type YqjM was codon optimized for expression in Escherichia coli and synthesized from GENWIZ (Suzhou, China). The gene YqjM was cloned into the plasmid pET28b(+) between Nde I and Hind III (N-terminal 6×His tag) and confirmed by sequencing. The obtaining plasmid pET28b(+)-YqjM was used as the template for mutagenesis. The YqjM mutants were generated using the KOD mutagenesis kit (TOYOBO, China) with the corresponding mutagenic primers (**Table S2**).” **was added.**

(10) Michaelis-Menten plot: Please provide more details on how the plot was obtained. For example, what are the actual substrate concentrations that were used in determining the initial rates (instead of just listing the range of 0-8 mM)? Are the initial rates obtained from just a single measurement at 10 min time point or are there in between time points? What program was used to fit the Michaelis-Menten plot into the obtained data? It is also strange to me that the Michaelis-Menten plots were presented as just the final fitted curves with no data points and error bars presented. These should be added, along with the final k_{cat} and K_M for each plot as well as the R^2 of the fit.

Response: The reviewer is absolutely right that our original description of the kinetic experiments was insufficient. We have improved the experimental section in the supporting information accordingly.

Modification: In the revised supporting information, Page 7, “

Enzymatic kinetic parameters

All kinetic experiments were conducted using a SpectraMax190 (Molecular devices, America) in PBS (100 mM, pH 7.5) and following the oxidation of NADPH at λ of 340 nm ($\epsilon = 6220 \text{ M}^{-1} \text{ cm}^{-1}$) at 30 °C for 10 min. The initial rates were obtained by measuring the absorbance change between time points, not just a single measurement at the 10-minute time point. For a standard activity assay, an assay reaction mixture (1 mL) contained 10 μM enzyme, a certain amount of substrate (0-15 mM; actual substrate concentrations see **Figure S11-S24**), 1 mM NADPH, and 1 mL PBS (100 mM, pH 7.5). The initial rates were calculated from the linear range of the fitted trend line of the progress curve. Kinetic parameters (k_{cat} and K_M) and respective standard errors were determined by measuring the activities at different substrate concentrations and fitting the activity versus substrate concentration data to the Michaelis-Menten equation using Origin 8.6.” **was added.**

Kinetics parameters of the YqjM mutants

WT

Figure S11 Michaelis-Menten-plots for the reduction of **1a** by WT YqjM.

Figure S12 Michaelis-Menten-plots for the reduction of **2a** by WT YqjM.

I69G

Figure S13 Michaelis-Menten-plots for the reduction of **1a** by YqjM (I69G).

Figure S14 Michaelis-Menten-plots for the reduction of **2a** by YqjM (I69G).

I69Y

Figure S15 Michaelis-Menten-plots for the reduction of **1a** by YqjM (I69Y).

Figure S16 Michaelis-Menten-plots for the reduction of **2a** by YqjM (I69Y).

I69Y/C26A

Figure S17 Michaelis-Menten-plots for the reduction of **1a** by YqjM (I69Y/C26A).

Figure S18 Michaelis-Menten-plots for the reduction of **2a** by YqjM (I69Y/C26A).

I69Y/C26G

Figure S19 Michaelis-Menten-plots for the reduction of **1a** by YqjM (I69Y/C26G).

Figure S20 Michaelis-Menten-plots for the reduction of **2a** by YqjM (I69Y/C26G).

I69Y/C26G/Y169F

Figure S21 Michaelis-Menten-plots for the reduction of **1a** by YqjM (I69Y/C26G/Y169F).

Figure S22 Michaelis-Menten-plots for the reduction of **2a** by YqjM (I69Y/C26G/Y169F).

I69Y/C26G/Y28F

Figure S23 Michaelis-Menten-plots for the reduction of **1a** by YqjM (I69Y/C26G/Y28F).

Figure S24 Michaelis-Menten-plots for the reduction of **2a** by YqjM (I69Y/C26G/Y28F).

REVIEWERS' COMMENTS

Reviewer #1 (Remarks to the Author):

The authors accomplished all corrections/modifications suggested in my first review of the manuscript. Manuscript and SI content are both notably improved and, in my opinion, are now ready for finalizing their publication in Nature Communications.

Reviewer #3 (Remarks to the Author):

All of my comments have been appropriately addressed by the authors in the revised manuscript. I believe that the revised manuscript can be published as is.